# Level and determinants of postpartum adherence to antiretroviral therapy in the Eastern Cape, South Africa

Oladele Vincent Adeniyi[1], Anthony Idowu Ajayi[2]*

**1** Department of Family Medicine & Rural Health, Faculty of Health Sciences, Walter Sisulu University, Mthatha/East London Hospital Complex, Cecilia Makiwane Hospital, East London, South Africa, **2** Population Dynamics and Sexual and Reproductive Health and Rights Unit, African Population and Health Research Center, APHRC Campus, Manga Close, Nairobi, Kenya

* ajayianthony@gmail.com, aajayi@aphrc.org

## Abstract

### Background

Adherence to antiretroviral therapy (ART), especially during the postpartum period, remains a major challenge in the efforts towards eliminating mother-to-child transmission of HIV. This study examined the levels and determinants of postpartum adherence to ART among mothers with HIV in the Eastern Cape, South Africa.

### Methods

In this cross-sectional analytical study, we interviewed 495 postpartum women with HIV between January and May 2018. We measured postpartum adherence using six questions probing participants' adherence behaviours since child birth. We categorised the adherence behaviours into complete adherence (mothers who reported no missed episode(s) of ART since child birth) and suboptimal adherence (mothers with any missed episode(s) of ART). Adjusted and unadjusted logistic regression models were used to examine the determinants of postpartum adherence to ART.

### Results

Overall, 63.9% reported complete adherence during the postpartum period but the rates varied by socio-demographic and behavioural characteristics. The adjusted logistic regression analysis showed that younger mothers were 70% less likely to report complete adherence to ART compared to mothers aged 40 and above. Likewise, mothers who currently use alcohol were 53% less likely to report complete postpartum adherence to ART compared to those who did not use alcohol. However, mothers who knew their partner's status were twice more likely to report complete postpartum adherence compared to those who did not. There was no statistically significant relationship between ART adherence and breastfeeding durations.

**Data Availability Statement:** All relevant data are within the manuscript and Supporting Information files.

**Funding:** OVA is a receipient of the South African Medical Research Council Pilot Grant awarded to

Walter Sisulu University. Research reported in this publication was supported by the South African AIDS Vaccine Initiative under the auspices of the South African Medical Research Council with funds received from the South African National Department of Health.

**Competing interests:** The authors have declared that no competing interests exist.

## Conclusion

Postpartum adherence to ART is suboptimal in the study setting, and younger mothers and those who use alcohol have a lower odds of complete adherence. Knowing a partner's status improves adherence, but infant feeding practices did not influence postpartum adherence behaviours. It is critical to design and strengthen interventions which target young mothers and alcohol users. Also, HIV sero-status disclosure should be encouraged among mothers to facilitate partner support.

## Introduction

The introduction of antiretroviral therapy (ART) has led to a significant reduction in new HIV infections and AIDS-related deaths globally [1]. AIDs-related deaths have declined by 52% since 2004, and the new HIV infection had fallen from a peak of 3.4 million in 1996 to 1.8 million in 2017 [2]. What is more, over 1.4 million new HIV infections have been averted among children since 2010 [2, 3]. Despite this progress, the reported number of people, particularly children, newly infected is still too high and clearly indicates that the war against HIV has not been won. ART has reduced in-utero transmission to less than 1%, but many babies are still getting infected during the postpartum period. Optimal and sustained (complete) adherence to ART is required for virological suppression, elimination of mother-to-child transmission of HIV, averting new horizontal HIV transmission, and preventing drug resistance and AIDS-related deaths [4, 5].

In 2016, the world health organization (WHO) recommended that mothers with HIV in developing countries should breastfeed exclusively for six months, introduce appropriate complementary feeding and continue to breastfeed for up to two years of the babies life [6]. However, the risk of mother to child transmission of HIV is 28% during the postnatal period but could reduce to 0.2% with the use of antiretroviral therapy (ART) [7]. Complete adherence to ART is required to achieve undetectable levels of breastmilk RNA, a condition necessary for the prevention of mother-to-child transmission (PMTCT) of HIV [4, 8]. Detectable breastmilk RNA is associated with elevated risk of breastmilk transmission [4]. However, complete postpartum adherence to ART is associated with a 52% reduction in the level of breast milk HIV transmission[8]. Even though sustaining viral suppression is crucial for PMTCT of HIV during the postnatal period, a study has shown that only half of women maintain viral suppression through 12 months postpartum in sub-Saharan Africa (SSA), a region with the highest number of people living HIV [5].

It appears that pregnancy motivates women to adhere to ART. Existing evidence suggests that adherence rate decline during the postnatal period after peaking during pregnancy [9–11]. Also, studies conducted in Uganda and Zambia suggest that adherence rate is higher among pregnant women compared to breastfeeding women [12–14]. Younger age, alcohol and tobacco use, higher parity, and postpartum depression are among the main factors associated with non-adherence to ART [15, 16], while disclosure and partner support facilitate adherence [17, 18]. With most mothers with HIV in SSA choosing to breastfeed [19], addressing postpartum non-adherence to ART is vital for realising the full potential of antiretroviral interventions to prevent mother-to-child transmission of HIV.

South Africa has the highest HIV epidemic in the world, with 19% of the global number of people with HIV, 15% of the new infection, and 11% of AIDS-related deaths [20]. Also, the country has the largest treatment programme accounting for 20% of all people on ART [20]. An estimated 13,000 children were newly infected due to MTCT in 2017, and many of these

occur during the postpartum period [20]. Adherence to ART remains a major challenge in eliminating mother-to-child transmission of HIV in the country despite the enormous government investment in ART [17]. Given that the risk of MTCT of HIV extends beyond pregnancy period and with breastfeeding mothers with poor adherence at risk of transmitting the virus to their suckling babies, it is crucial to examine adherence to ART beyond the pregnancy period. It is evident that advancing effective strategies to tackle non-adherence to ART is required to eliminate new HIV infection, especially mother-to-child transmission of HIV in South Africa

Most studies on adherence to ART in sub-Saharan Africa tend to focus on pregnant women. Also, existing studies have shown that adherence to ART is higher among pregnant women compared to postpartum mothers. While we know that adherence to ART declines during the postpartum period, it remains unclear how infant feeding practices influence adherence to ART. Based on a previous study that shows that women with older children, most of whom stopped breastfeeding by 13–18 months, reported more barriers and missed doses than women with younger children [21], we assumed in this study that breastfeeding duration may influence adherence to ART. Also, knowing partner's status could create an enabling environment that increases support for adherence; however, the relationship between knowing a partner's status and postpartum adherence has not been explored in the Eastern Cape, South Africa, a resource-constrained province in South Africa with the third largest HIV prevalence in the country. In this study, we examine the level of postpartum adherence to ART. Also, we examined the influence of infant feeding practices, younger age, knowing partner's status and alcohol use on postpartum adherence to ART.

## Methods

### Study design and settings

To understand the level of postpartum adherence, we conducted a cross-sectional survey of parturient women with HIV enrolled in the electronic database of the East London Prospective Cohort Study after 18–24 months post-delivery. This electronic database was created specifically for research purposes to track the PMTCT outcomes of parturient women with HIV who delivered at the three largest maternity facilities between September 2015 and May 2016 in the Buffalo City and Amathole districts of the East Cape province of South Africa. These maternity centres serve approximately 1.7 million people from rural, peri-urban and urban communities of the selected districts in the central region of the Eastern Cape. The detailed methodology of the East London Prospective Cohort Study has been published elsewhere [22].

### Participants and procedure

Postpartum women enrolled in the East London Prospective Cohort Study database were contacted telephonically to participate in the exit survey between January to May 2018. Participants were given options to either complete interviewer-guided interview in person in one of the three hospitals used for the baseline study or on the phone. The research team reimbursed the respondents for the cost of transportation to the hospital. Those who were unable to come to the hospital agreed to complete the survey questions via telephone interview at their most convenient time. We recruited research assistants who understood the IsiXhosa language (local language) and trained them specifically for this study. The research assistants administered the study questionnaires. We successfully contacted 509 participants. Some of the eligible participants could not be reached through their mobile phones or on any of the three contactable numbers obtained at the baseline. We estimated, using the Cochrane formula for estimating sample size for categorical data, a sample size of 485 at a confidence level of 95%, a precision level of +/−4% and 10% possible attrition.

## Measures

The questionnaire consisted of relevant items on socio-demographic factors, adherence to current ART and behavioural lifestyles. The main outcome of interest in this study is postpartum adherence to ART medications. We measured postpartum adherence using six questions, probing participants' adherence behaviours since child birth. The measure of adherence in this study is focused on complete daily episodes of consistent use of ART and not on hourly variations in the use of ART. These questions have a "yes" or "no" response. The six items yielded a reliability score (Cronbach's Alpha coefficient) of 0.69, which indicate high internal consistency. Our measure of adherence, based on self reporting, has been demonstrated to be robust and predictive of virological suppression [14, 23] and has been previous validated [24]. A response of yes was coded as "1" and no as "0". A final score between 1–6 was considered to be suboptimal adherence, while a score of 0 indicates complete adherence. A score of zero means participants affirmed to have regularly used their medication since the birth of their child irrespective of all scenarios indicated in our six questions. A score between one and six means that there are instances when respondents have not used their medications.

The main covariates included in this study were demographic and behavioural factors. Age of the respondents was obtained and coded as a continuous variable. Employment and marital status, as well as the level of education, were also obtained. The behavioural factors include smoking and alcohol use, which were categorical variables with "yes" or "no" responses. Further, participants were asked whether they had disclosed their status to their partner and if they knew their partner's HIV serostatus. Answers to these questions were binary (yes/no). Infant feeding practice was obtained, including formula feeding, duration of breastfeeding, and breastfeeding cessation.

## Ethical approval and consent to participate

The Walter Sisulu University Ethical Review Committee granted approval for the study protocol (Reference: 085/2017). Also, permission to conduct the study was obtained from the Eastern Cape Department of Health and the management of the Cecilia Makiwane Hospital. All the participants in the baseline study signed written informed consent for their readiness to participate in the follow-up study. Additional informed consent was obtained from each participant either verbally (telephonic survey) or in writing (face-to-face interview) after they affirmed that they understood their right to participate in the study freely, and to refuse to answer any question they are not comfortable with or even drop out at any time. The rights of participants to privacy and confidentiality of medical information were respected throughout the study.

## Statistical analysis

Descriptive statistics, including mean, frequency count and percentages, were used to summarise the data. We fitted adjusted and unadjusted logistic regression models to examine factors associated with complete adherence to ART during the postpartum period. 95% confidence intervals were reported for all analyses, and a p-value less than 0.05 was considered statistically significant. All analyses were conducted with the Statistical Packages for Social Sciences, version 24.0 (SPSS, Chicago, IL, USA).

## Results

### Socio-demographic characteristics

The analysis was performed on 485 participants who answered the questions on ART adherence. Of the 509 participants contacted, nine had died, and six refused to participate in the

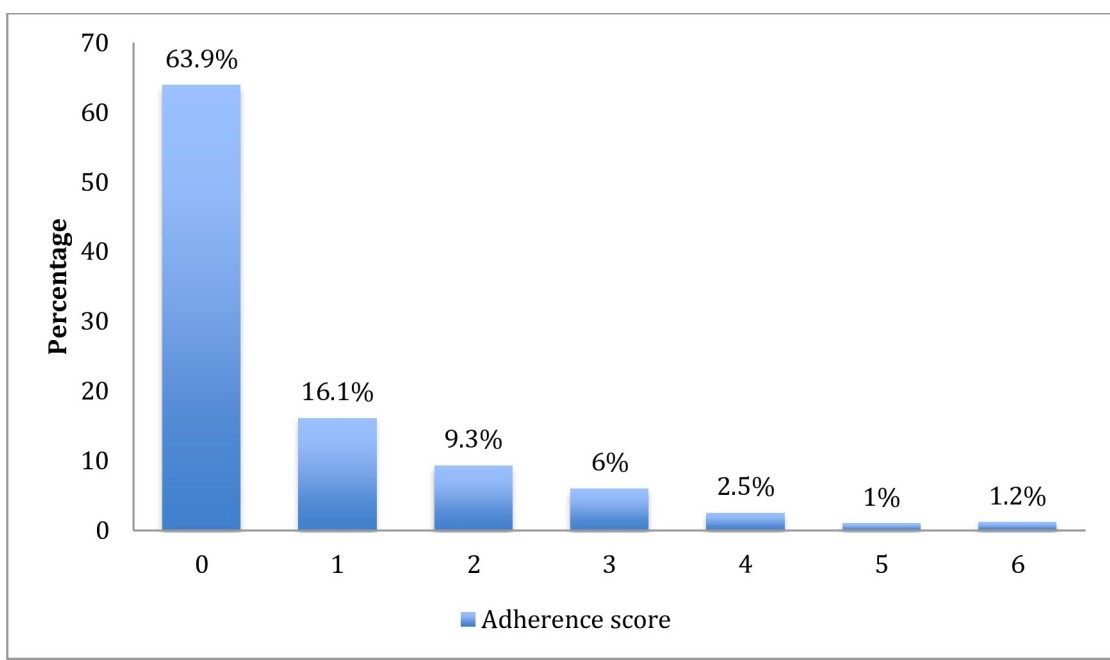

**Fig 1. Distribution of participants by adherence score.**

study, while nine did not answer the questions on adherence to ART, having withdrawn their participation from the study. The mean age of study participants was 32.91 (SD 5.74) years. Most participants were aged 25 years or more (92.6%), single (67%), had grade 12 education, unemployed (62.3%), received government's social grant (93.6%), non-smokers, and never drank alcohol (64.3%). The mean duration of HIV infection was 6.7 years (SD = 4.1 years).

### Descriptive findings

The overall postpartum adherence rate was 63.9% but varied by socio-demographic and behavioural characteristics. The distribution of adherence scores is presented in Fig 1. Adherence was lowest among women aged 24 years or less (36.1%), students (43.5%), women cohabiting (46.2%), alcohol users (50.9%) and smokers (51.2%). However, a higher level of adherence was reported among women who had disclosed their status to their partner (65.8%) and those who knew their partner's status (69.0%). There was no significant difference in the level of adherence by infant feeding choices (Table 1).

### Multivariable analyses

The results of the unadjusted logistic regression models show that younger age and alcohol use were associated with lower odds of postpartum adherence to ART. While knowing a partner's status was associated with a higher likelihood of postpartum adherence to ART. However, infant feeding practices were not associated with postpartum adherence. The magnitude and direction of effects remain in the adjusted regression (Table 2). In the adjusted logistic regression, younger mothers were 70% less likely to report complete adherence to ART. Likewise, mothers who currently use alcohol were 53% less likely to report complete postpartum adherence to ART. However, mothers who knew their partner's status were twice more likely to report complete postpartum adherence (AOR: 2.07; CI:1.25–3.41). Shorter duration of breastfeeding was not significantly associated with lower odds of complete adherence to ART.

**Table 1. Chi-square statistics showing factors associated with complete adherence to ART.**

| Variables | All participants | Complete adherence | Sub-optimal adherence | p-value |
|---|---|---|---|---|
| Age | | | | |
| 24 years or less | 36 (7.4) | 13 (36.1) | 23 (63.9) | 0.003 |
| 25–29 years | 114 (23.5) | 73 (64.0) | 41 (36.0) | |
| 30–34 years | 144 (29.7) | 89 (61.8) | 55 (38.2) | |
| 35–39 years | 123 (25.4) | 88 (71.5) | 35 (28.5) | |
| 40 years and above | 68 (14.0) | 47 (69.1) | 21 (30.9) | |
| Marital status | | | | |
| Never married | 325 (67.0) | 202 (62.2) | 123 (37.8) | 0.039 |
| Currently married | 126 (26.0) | 89 (70.6) | 37 (29.4) | |
| Cohabiting | 26 (5.4) | 12 (46.2) | 14 (53.8) | |
| Previously married | 8 (1.6) | 7 (87.5) | 1 (12.5) | |
| Education level | | | | |
| Grade 7 and less | 29 (6.0) | 22 (75.9) | 7 (24.1) | 0.220 |
| Grade 8–12 | 421 (86.8) | 263 (62.5) | 158 (37.5) | |
| Higher education | 34 (7.2) | 25 (71.4) | 10 (28.6) | |
| Employed in a salary paying job | | | | |
| Yes | 157 (32.4) | 100 (63.7) | 57 (36.3) | 0.511 |
| No | 328 (67.6) | 210 (64.0) | 118 (36.0) | |
| Occupation in last 12 months | | | | |
| Government employee | 17 (3.5) | 11 (64.7) | 6 (35.3) | 0.155 |
| Non-government employee | 114 (23.5) | 80 (70.2) | 34 (29.8) | |
| Self employed | 29 (6.0) | 20 (69.0) | 9 (31.0) | |
| Student | 23 (4.7) | 10 (43.5) | 13 (56.5) | |
| Unemployed | 302 (62.3) | 189 (62.6) | 113 (37.4) | |
| Smoking | | | | |
| Yes | 43 (8.9) | 22 (51.2) | 21 (48.8) | 0.050 |
| No | 442 (91.1) | 288 (65.2) | 154 (34.8) | |
| Drank alcohol in the past year | | | | |
| Yes | 173 (35.7) | 88 (50.9) | 85 (29.1) | <0.001 |
| No | 312 (64.3) | 222 (71.2) | 90 (28.8) | |
| Know partner's status | | | | |
| Yes | 319 (65.8) | 220 (69.0) | 99 (31.0) | 0.001 |
| No | 166 (34.2) | 85 (53.8) | 73 (46.2) | |
| Disclosed status to partner | | | | |
| Yes | 400 (82.5) | 263 (65.8) | 137 (34.3) | 0.027 |
| No | 85 (17.5) | 44 (53.7) | 38 (46.3) | |
| Breastfeeding duration | | | | |
| 0 month | 187 (38.6) | 121(64.7) | 66 (35.3) | 0.942 |
| 1–3 months | 105 (21.6) | 64 (61.0) | 41 (39.0) | |
| 4–6 months | 116 (23.9) | 74 (63.8) | 42 (36.2) | |
| 7–12 months | 44 (9.1) | 28 (63.6) | 16 (36.4) | |
| 12 months and more | 32 (6.6) | 22 (68.8) | 10 (31.3) | |
| Number of years since HIV diagnosis | | | | |
| 1–5 years | 206 (42.5) | 124 (60.2) | 82 (39.8) | 0.312 |
| 6–10 years | 163 (33.6) | 107 (65.6) | 56 (34.4) | |
| 11–17 years | 116 (23.9) | 79 (68.1) | 37 (31.9) | |

**Table 2. Adjusted and unadjusted binary logistic regression showing determinants of complete adherence to ART.**

| Variables | Unadjusted odds | Adjusted Odds |
|---|---|---|
| Age | | |
| 24 years and less | 0.25 (0.11–0.59)* | 0.30 (0.12–0.77)* |
| 25–29 years | 0.80 (0.42–1.51) | 0.85 (0.42–1.64) |
| 30–34 years | 0.72 (0.39–1.34) | 0.70 (0.36–1.34) |
| 35–39 years | 1.12 (0.59–2.15) | 1.09 (0.55–2.15) |
| 40 years and above | 1 | 1 |
| Marital status | | |
| Currently married | 1.50 (0.97–2.33) | 1.13 (0.70–1.83) |
| Never married or previously married | 1 | 1 |
| Education level | | |
| Grade 7 and less | 1.26 (0.41–3.87) | 1.12 (0.33–3.77) |
| Grade 8–12 | 0.67 (0.31–1.42) | 0.81 (0.36–1.86) |
| Higher education | 1 | 1 |
| Occupation in the past 12 months | | |
| Employed | 1.35 (0.90–2.04) | 1.53 (0.98–2.39) |
| Student | 0.46 (0.20–1.08) | 0.65 (0.25–1.73) |
| Unemployed | 1 | 1 |
| Smoking status | | |
| Yes | 0.56 (0.30–1.05) | 0.76 (0.38–1.53) |
| No | 1 | 1 |
| Drank alcohol in the past year | | |
| Yes | 0.42 (0.29–0.62)*** | 0.47 (0.31–0.72)*** |
| No | 1 | 1 |
| Disclosure of status to a partner | | |
| Yes | 1.55 (0.97–2.50) | 0.89 (0.48–1.66) |
| No | 1 | 1 |
| Knows partner's HIV status | | |
| Yes | 1.88 (1.28–2.76)* | 2.07 (1.25–3.41)* |
| No | 1 | 1 |
| Breastfeeding duration | | |
| 0 month | 0.83 (0.37–1.87) | 0.90 (0.38–2.12) |
| 1–3 months | 0.71 (0.31–1.65) | 0.71 (0.29–1.75) |
| 4–6 months | 0.80 (0.35–1.85) | 0.79 (0.32–1.91) |
| 7–12 months | 0.80 (0.30–2.09) | 0.79 (0.28–2.20) |
| 12 months and more | 1 | 1 |

*$p$-value <0.05

***$p$-value <0.001

## Discussion

The study examined the level of and factors associated with postpartum adherence to ART in the Eastern Cape, South Africa. Our analysis shows that fewer than two-thirds of mothers self-reported complete adherence to ART at 18–24 months postpartum period. Even though a 100% rate of postpartum adherence is desirable, the 64% adherence level in this study is higher than the rate reported among postpartum mothers in a study in western Uganda [12] and in SSA [5]. Of concern, however, is the decline in the level of adherence among these women

relative to the level reported during pregnancy [17]. The decline is even more pronounced among adolescent and young adults, with less than 37% adherence rate compared to 63.5% rate during pregnancy [17]. Compared to women aged 40 years or more, adolescents and young adults had a 70% lower odds of attaining complete adherence to ART. The finding is not surprising given that younger age has been reported previously to be associated with non-adherence to ART [15]. Thus, any intervention to improve postpartum adherence must prioritise adolescents and young adults in the study settings.

While we assumed that breastfeeding duration will positively influence adherence to ART among postpartum mothers with HIV, our analysis shows that the association is not statistically significant. In contrast to a study that links breastfeeding cessation with adherence challenges [21], our study did not find a significant difference in the level of adherence between mothers that did not initiate breastfeeding, breastfeed for a short duration or those who breastfed for a longer duration. Fear of infecting the baby is one of the main reasons for not initiating breastfeeding [17], and the group of women who did not breastfeed are likely to take adherence seriously, given their concern for not transmitting the virus. Likewise, mothers that breastfeed their babies knew the importance of adherence to ART to prevention of MTCT of HIV and are likely to use their medication to reduce their risk of transmitting HIV to their infants. Another plausible explanation could be that since South Africa is a largely breastfeeding country, with breastfeeding being practiced religiously in many rural communities [19], it is likely that the there is a uniform distribution of breastfeeding across communities and may, therefore, not be a good predictor of adherence.

Our study also shows that alcohol use significantly influenced postpartum adherence to ART, which is consistent with previous studies [17, 25, 26]. Alcohol users were fifty-three percent less likely to adhere to ART compared to non-users. The pathway through which alcohol use interfere with adherence to ART in the study setting has been explained in a previous study [17]. Alcohol users tend to drink in company of friends on nights out and in parties, which makes them forget to use their medications [17]. Also, it appears that people believe that they can not use their medication as well as take alcohol making them to sometimes forgo ART in order to take alcohol [25, 26]. Young people are more likely to use alcohol compared to adults [27], and this may explain why adherence is poor among young women compared to older adults. Screening for alcohol use may help to identify mothers with adherence challenges who might require adherence counselling during postnatal care and child immunisation.

Another important finding of this study is that knowing partner's serostatus predicts postpartum ART adherence. Mothers who knew the status of their partners were twice more likely to adhere to ART compared to those who did not. Open and honest communication about HIV status can create environments conducive for adherence, with couples encouraging and supporting each other to achieve complete adherence. In such open and honest environment, there is a high tendency for the men to support their partners in ensuring complete adherence to ART in order to eliminate the risk of MTCT of HIV, a goal shared by both parents. In cases where both partners are not honest about their status, fear of stigma would hinder adherence to ART [17, 21].

## Study limitations

This study has some limitations. The association reported in this study could not be interpreted as causation given the cross-sectional nature of the data. Also, our measure of adherence is based on self-reporting; thus, adherence level may have been exaggerated due to social desirability bias. Nevertheless, the use of multiple questions to assess adherence offers a more reliable approach to measuring this concept as demonstrated in previous studies [14, 28, 29].

## Conclusion

Overall, postpartum adherence to ART is suboptimal in the study setting. Younger mothers and those who use alcohol have lower odds of reporting good adherence to ART. Knowing a partner's status improves adherence, but infant feeding practices did not influence postpartum adherence behaviours. This study has re-iterated factors associated with adherence to ART which have been reported in different contexts within and outside South Africa. It is critical to design and strengthen interventions which target young mothers and alcohol users. Also, HIV sero-status disclosure should be encouraged among mothers to facilitate partner support.

## Supporting information

**S1 File. Study dataset.**
(SAV)

## Acknowledgments

The authors are greatful to the study participants for taking their time to share their experience and personal information for this project. Also we appreciate our amazing research assistants for their contribution during the data collection phase of this project.

## Author Contributions

**Conceptualization:** Oladele Vincent Adeniyi, Anthony Idowu Ajayi.

**Data curation:** Oladele Vincent Adeniyi, Anthony Idowu Ajayi.

**Formal analysis:** Anthony Idowu Ajayi.

**Funding acquisition:** Oladele Vincent Adeniyi.

**Investigation:** Oladele Vincent Adeniyi, Anthony Idowu Ajayi.

**Methodology:** Oladele Vincent Adeniyi, Anthony Idowu Ajayi.

**Project administration:** Oladele Vincent Adeniyi, Anthony Idowu Ajayi.

**Resources:** Oladele Vincent Adeniyi.

**Supervision:** Oladele Vincent Adeniyi, Anthony Idowu Ajayi.

**Validation:** Oladele Vincent Adeniyi.

**Writing – original draft:** Oladele Vincent Adeniyi, Anthony Idowu Ajayi.

**Writing – review & editing:** Oladele Vincent Adeniyi, Anthony Idowu Ajayi.

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
