## [Decision Letter · Decision Letter 0]

20 Dec 2019

PONE-D-19-31896

Levels and determinants of postpartum adherence to Antiretroviral Therapy in the Eastern Cape, South Africa

PLOS ONE

Dear Dr Ajayi,

Thank you for submitting your manuscript to PLOS ONE. After careful consideration, we feel that it has merit but does not fully meet PLOS ONE’s publication criteria as it currently stands. Therefore, we invite you to submit a revised version of the manuscript that addresses the points raised during the review process.

The reviewers note that this paper reports on an important outcome namely postpartum ART adherence. They note some changes that would strengthen the paper. Please respond to these comments and revise accordingly especially regarding how you defined adherence and how the adherence data was analysed and also address the issue of selection bias.

We would appreciate receiving your revised manuscript by Feb 02 2020 11:59PM. To enhance the reproducibility of your results, we recommend that if applicable you deposit your laboratory protocols in protocols.io, where a protocol can be assigned its own identifier (DOI) such that it can be cited independently in the future. For instructions see: http://journals.plos.org/plosone/s/submission-guidelines#loc-laboratory-protocols

We look forward to receiving your revised manuscript.

Kind regards,

Tanya Doherty, PhD

Academic Editor

PLOS ONE

Journal Requirements:

2. Please include additional information regarding the survey or questionnaire used in the study and ensure that you have provided sufficient details that others could replicate the analyses. If you developed and/or translated a questionnaire as part of this study and it is not under a copyright more restrictive than CC-BY, please include a copy, in both the original language and English, as Supporting Information.

4. Your ethics statement must appear in the Methods section of your manuscript. If your ethics statement is written in any section besides the Methods, please move it to the Methods section and delete it from any other section. Please also ensure that your ethics statement is included in your manuscript, as the ethics section of your online submission will not be published alongside your manuscript.

Reviewers' comments:

Reviewer's Responses to Questions

**Comments to the Author**

1. Is the manuscript technically sound, and do the data support the conclusions?

Reviewer #1: Yes

Reviewer #2: Yes

2. Has the statistical analysis been performed appropriately and rigorously? 

Reviewer #1: Yes

Reviewer #2: Yes

3. Have the authors made all data underlying the findings in their manuscript fully available?

Reviewer #1: No

Reviewer #2: No

4. Is the manuscript presented in an intelligible fashion and written in standard English?

Reviewer #1: Yes

Reviewer #2: Yes

5. Review Comments to the Author

Reviewer #1: Summary:

This is a very well written paper and easy to read and understand. It presents an important long-standing issue of attempting to understand the reasons behind suboptimal adherence to antiretroviral therapy (ART) among HIV-positive postnatal women. They find that in the context of a community of the Eastern Cape province of South Africa, three factors are associated with ART adherence during the postnatal period. Young maternal age and alcohol use are associated with poor adherence while knowledge of partner’s HIV status is associated with better adherence. These associations are commonly observed in the rest of the country and other parts of Sub-Saharan Africa. However, the authors were mostly interested in how infant feeding practices influence adherence and they did not find a relationship between ART adherence and infant feeding practices.

Their results probably reflect that even though certain factors are commonly associated with ART adherence and should be addressed with confidence at a population level, the outcome related to the hypothesis behind exploring the influence of infant feeding practices on ART adherence could be masked by the local community’s societal and cultural influences. Future work incorporating perceptions around breastfeeding and HIV infection would be useful.

Although it is a brilliant study, the authors may consider looking into a few suggestions which could make the manuscript stronger. The suggestions are not major but do require the authors’ attention.

Minor Essential Revisions:

1. ABSTRACT – probably mention in a short sentence what method was used to measure adherence.

2. ABSTRACT: Be careful how you interpret results about feeding. There is no indication of statistical significance and the confidence intervals fully overlap a statistic of no significant (OR=0). Therefore, I would recommend you clearly state that there was not relationship between ART adherence and feeding rather than saying feeding ‘tended to reduce the odds of adherence’.

3. STATISTICAL METHODS AND RESULTS:

a. The way you measured adherence appears to be a strong and robust method but however the way you present the results has masked this robust approach. Your method of measuring adherence appears to report an ordered outcome variable with scores ranging from ‘0’ through to ‘6’. An important question is whether scoring a 1 reflects better adherence challenges compared to scoring a 6, on average? If this is true, then your outcome should be ordered from perfect adherence (score=0) to poorest/worst (score=6). It would make sense to have four ordered groups with adherences scores of 0, 1-2, 3-4 and 5-6. You would then check if the ordered logit model fits your data better than your current logistic regression model with a binary outcome (score 0 versus 1-6). You might need to include a supplementary file explaining how you decided the choose the final approach.

b. You need to include a distribution of the adherence scale results for the overall sample, probably in the form of a graph, in the main body of the results section. You mention that ~64% of participants adhered (abstract and discussion), but the reader does not see this clearly from the results section! Did 64% get an adherence score of 0? It is important for the reader to see what proportion of the sample fell into the different scores of the adherence scale 0, 1, 2, 3, 4, 5, 6 or stepwise groups 0, 1-2, 3-4, 5-6. This will also help the reader to understand why you chose a binary outcome over an ordinal outcome or vice-versa. Please add a paragraph describing the main outcome variable in the results section.

c. Under 'Postpartum Adherence to ART' paragraph, you mention that adherence was also lowest among 'students (43.2%)'. Where is this outcome in Table 2?

d. It will be useful to mention what statistic was done to get the p-value reported in Table 2. Yoy may add a table legend with this information.

4. DISCUSSION: In the paragraph which talks about feeding practices, consider discussing the reasons why this dataset did not reflect your hypotheses around feeding and ART adherence? Have you considered the possibility of socio-cultural perceptions around breastfeeding and the stigma associated with HIV if a woman does not breastfeed, in certain settings. If this is true, then breastfeeding will likely be more uniform across the community thus not be a good predictor of adherence. South Africa is largely a breastfeeding country and many rural communities practice breastfeeding religiously.

5. DISCUSSION: You mention that ‘young people are more likely to use alcohol ….’. This statement requires a qualification either from an external reference or from your dataset. Does your data show an association between young age and alcohol use? Otherwise I would not recommend the statement.

6. CONCLUSION: This study has re-iterated associations which have been reported in different contexts within and outside South Africa. Hence emphasis is needed on this and that it is time to focus on designing or strengthening interventions which deal with the risk of young maternal age and alcohol.

Reviewer #2: The study documents a cross sectional analysis of adherence to ARVs and determinants thereof among postpartum mothers with HIV who delivered at a hospital in the Eastern Cape, South Africa. Data collection takes place 18-24 months postpartum and this study appears to be one part of a larger study/ies based on the cohort “The East London Prospective Cohort Study after 18-24 months post-delivery”, although how this analysis fits in with this cohort and any other studies, if at all, is unclear

There are no line numbers, which makes it difficult to be clear about the recommended edits.

Overall: The study needs to be published as the information is useful to health practitioners and policy makers, and will be of interest to researchers. However the quality of the writing needs to be strengthened for publication in PLoS One

Abstract: Give a brief definition of adherence vs perfect adherence – see my point fi Introduction and be consistent throughout

Introduction

1. Please give a succinct account of what perfect adherence involves – daily medications at a specific time – or what – or is there a variation

2. Para 2, 3 and 4: Please can you attempt to rewrite with a more crisp style. Avoid words like “huge”. Also give dates for your statistics

3. Para5: “The Eastern Cape is a resource constrained… “- please correct

Methods

Design and setting

1. Please clarify how this analysis relates to the electronic database – is the database routine surveillance, or research driven. Don’t just refer to other literature.

2. 18-24 months is quite a long time – in the methods and in the results please specify how many months or weeks post delivery the participants were, and if a range – do a bivariate analysis on adherence and if this is significant, include in the MV analysis

Participants

3. Please clarify whether all women in the database were contacted to participate or a sub-sample. If a sub-sample – how were they selected?. If not able to be contacted, what are the characteristics of those who could not be contacted. Please address the question of selection bias thoroughly.

4. There is no sample size calculation – please include or explain why not

Measures

5. Rather than describing the questions, please include them verbatim. Also, I counted only 5 question descriptions .

6. Have the questions been validated against other measure of adherence that are not self-report or are they ‘de novo’– please give details and explanations

Statistical analysis

7. The first 2 sentences fit better into the results. Also include characteristics of those not included in the 509 as well as the 10/509 who did not answer the adherence questions – and give reasons why they didn’t

Para 2

8. Clarify the outcome – “perfect adherence”

9. Typo “A 95% confidence intervals..” – please correct

Results

1. Para 1“Receives” s/b received

2. Please add HIV duration and categories of months/years duration of HIV infection eg. 1.5-2yrs, 2-5 yrs, >5 - or some other meaningful division and include in the Tables , and comment on the result in the discussion. The same for months postpartum

3. Only keep this text if Table 1 is removed. Otherwise it is just a repeat

4. Postpartum adherence: Specify that you mean the overall perfect adherence rate in sentence 1 and thereafter wherever this is the case. Or define adherence earlier in the text to mean perfect and the be consistent throughout the paper only using the word adherence

5. Multivariable findings: Change to Multivariable analyses

6. Remove the word “at least” as it is inaccurate, and cite the 95% CI

Tables

1. Table 1 is unnecessary as the information can be incorporated as a column with n (%) in

2. Table 2: It would strengthen the paper if you divide suboptimal adherence into at least 2 categories – based on the score. The ORs can compare perfect to suboptimal adherence overall, but this additional information is worth informing the reader

3. Add footnotes to Table 3 to explain the *, **, ***

Discussion

1. Para 1 last sentence doesn’t make sense if you include “likely”. Please be specific

2. Para 2.Explain why you assumed breast feeding will influence adherence – perhaps in the introduction and refer back to that with clear reasoning

3. Par 3: Within this paragraph, you give details regarding decline – why not give these in Para1 and link these ideas

4. Study limitations: Replace sentence 1 with “This study has some limitations”

References:

1. Reference 17 is confusing – last author JJBid? no journal stated

6. PLOS authors have the option to publish the peer review history of their article (what does this mean?). If published, this will include your full peer review and any attached files.

Reviewer #1: No

Reviewer #2: Yes: Deborah Constant

---

## [Author Response · Author response to Decision Letter 0]

15 Jan 2020

PONE-D-19-31896

Levels and determinants of postpartum adherence to Antiretroviral Therapy in the Eastern Cape, South Africa

PLOS ONE

Dear Editor, 

Thank you for considering our article and returning constructive comments that have helped us to further improve our paper. We have studied the comments and have responded to all comments below. We trust you will find our manuscript good enough for publication. 

Best Regards

Anthony

Comments to the Author

Reviewer #1: Summary:

This is a very well written paper and easy to read and understand. It presents an important long-standing issue of attempting to understand the reasons behind suboptimal adherence to antiretroviral therapy (ART) among HIV-positive postnatal women. They find that in the context of a community of the Eastern Cape province of South Africa, three factors are associated with ART adherence during the postnatal period. Young maternal age and alcohol use are associated with poor adherence while knowledge of partner’s HIV status is associated with better adherence. These associations are commonly observed in the rest of the country and other parts of Sub-Saharan Africa. However, the authors were mostly interested in how infant feeding practices influence adherence and they did not find a relationship between ART adherence and infant feeding practices.

Their results probably reflect that even though certain factors are commonly associated with ART adherence and should be addressed with confidence at a population level, the outcome related to the hypothesis behind exploring the influence of infant feeding practices on ART adherence could be masked by the local community’s societal and cultural influences. Future work incorporating perceptions around breastfeeding and HIV infection would be useful.

Response

We thank the reviewer for the positive feedback.

Although it is a brilliant study, the authors may consider looking into a few suggestions which could make the manuscript stronger. The suggestions are not major but do require the authors’ attention.

Response

Many thanks to the reviewer for the important insights and comments. 

Minor Essential Revisions:

1. ABSTRACT – probably mention in a short sentence what method was used to measure adherence.

Response

We have added a sentence on adherence measure in the abstract.

2. ABSTRACT: Be careful how you interpret results about feeding. There is no indication of statistical significance and the confidence intervals fully overlap a statistic of no significant (OR=0). Therefore, I would recommend you clearly state that there was not relationship between ART adherence and feeding rather than saying feeding ‘tended to reduce the odds of adherence’.

Response

We have effected this correction.

3. STATISTICAL METHODS AND RESULTS:

a. The way you measured adherence appears to be a strong and robust method but however the way you present the results has masked this robust approach. Your method of measuring adherence appears to report an ordered outcome variable with scores ranging from ‘0’ through to ‘6’. An important question is whether scoring a 1 reflects better adherence challenges compared to scoring a 6, on average? If this is true, then your outcome should be ordered from perfect adherence (score=0) to poorest/worst (score=6). It would make sense to have four ordered groups with adherences scores of 0, 1-2, 3-4 and 5-6. You would then check if the ordered logit model fits your data better than your current logistic regression model with a binary outcome (score 0 versus 1-6). You might need to include a supplementary file explaining how you decided to choose the final approach.

Response

Our goal is to determine the rate and determinants of complete adherence since child delivery. Our use of six-item questions focusing on whether mothers have defaulted in using their medication since child delivery is to ensure a close to accurate measure of adherence. A score of six may not necessarily mean more adherence challenge given that the content of the six questions. Put differently, a person who reported any instances of not using their medication, may or may not use the medication in the past month, or may or may not use the medication because of alcohol or travel. So, a score of 1 or 6 may not necesarily indicate more adherence challenges. Since our interest is in complete adherence, only a score of zero, those with complete adherence, is of interest to us. It is critical to ensure complete adherence among parturient women, especially breastfeeding population to ascertain that they adhere to the current ART. We have however presented the scores and explain why we favoured a binary classification over an ordinal classification. We fitted ordered logit model and the variables that were significant in the logistic regression model were all significant in the ordered logit model. 

b. You need to include a distribution of the adherence scale results for the overall sample, probably in the form of a graph, in the main body of the results section. You mention that ~64% of participants adhered (abstract and discussion), but the reader does not see this clearly from the results section! Did 64% get an adherence score of 0? It is important for the reader to see what proportion of the sample fell into the different scores of the adherence scale 0, 1, 2, 3, 4, 5, 6 or stepwise groups 0, 1-2, 3-4, 5-6. This will also help the reader to understand why you chose a binary outcome over an ordinal outcome or vice-versa. Please add a paragraph describing the main outcome variable in the results section.

Response

This is an important suggestion and we have now added this as suggested.

c. Under 'Postpartum Adherence to ART' paragraph, you mention that adherence was also lowest among 'students (43.2%)'. Where is this outcome in Table 2?

Response

It is in Table under occupation, but it is 43.5% as we stated.

d. It will be useful to mention what statistic was done to get the p-value reported in Table 2. You may add a table legend with this information.

Response

We have added the statistical test from which the P-values were estimated.

4. DISCUSSION: In the paragraph which talks about feeding practices, consider discussing the reasons why this dataset did not reflect your hypotheses around feeding and ART adherence? Have you considered the possibility of socio-cultural perceptions around breastfeeding and the stigma associated with HIV if a woman does not breastfeed, in certain settings. If this is true, then breastfeeding will likely be more uniform across the community thus not be a good predictor of adherence. South Africa is largely a breastfeeding country and many rural communities practice breastfeeding religiously.

Response

We thank the reviewer for this suggestion. We did not consider this alternative explanation and have now added this. 

5. DISCUSSION: You mention that ‘young people are more likely to use alcohol ….’. This statement requires a qualification either from an external reference or from your dataset. Does your data show an association between young age and alcohol use? Otherwise I would not recommend the statement.

Response 

We have added references to the statement. 

6. CONCLUSION: This study has re-iterated associations which have been reported in different contexts within and outside South Africa. Hence emphasis is needed on this and that it is time to focus on designing or strengthening interventions which deal with the risk of young maternal age and alcohol.

Response

We thank the reviewer for this important suggestion

Reviewer #2: The study documents a cross sectional analysis of adherence to ARVs and determinants thereof among postpartum mothers with HIV who delivered at a hospital in the Eastern Cape, South Africa. Data collection takes place 18-24 months postpartum and this study appears to be one part of a larger study/ies based on the cohort “The East London Prospective Cohort Study after 18-24 months post-delivery”, although how this analysis fits in with this cohort and any other studies, if at all, is unclear

There are no line numbers, which makes it difficult to be clear about the recommended edits.

Response

We have added line numbers.

Overall: The study needs to be published as the information is useful to health practitioners and policy makers, and will be of interest to researchers. However the quality of the writing needs to be strengthened for publication in PLoS One

Response

We thank the reviewer for the positive review of our paper and suggestions provided to help improve our paper.

Abstract: Give a brief definition of adherence vs perfect adherence – see my point fi Introduction and be consistent throughout

Response

We have now added a definition of complete adherence and sub-optimal adherence. For consistency, we have changed the word “perfect adherence” to complete adherence all through the manuscript. Our use of adherence reflects complete adherence to ART. 

Introduction

1. Please give a succinct account of what perfect adherence involves – daily medications at a specific time – or what – or is there a variation

Response

It involves the use of ART regularly without any missing episodes. Our focus is not on hourly variation on a particular day, rather on days patients failed to use the medications. We have clarified this in the introduction.

2. Para 2, 3 and 4: Please can you attempt to rewrite with a more crisp style. Avoid words like “huge”. Also give dates for your statistics

Response

We have added dates to the statistics and rephrased the sentences where the word “huge” was used.

3. Para5: “The Eastern Cape is a resource constrained… “- please correct

Response 

Thank you for calling our attention to this. We have corrected it.

Methods

Design and setting

1. Please clarify how this analysis relates to the electronic database – is the database routine surveillance, or research driven. Don’t just refer to other literature.

Response

We thank the reviewer for this comment. We have indicated this database is not routine but research driven. The database contains information of mothers who gave birth between September 2015 to May 2016 period only. 

2. 18-24 months is quite a long time – in the methods and in the results please specify how many months or weeks post delivery the participants were, and if a range – do a bivariate analysis on adherence and if this is significant, include in the MV analysis

Response

We have specified that the months and weeks the participants were. It is a range of 18 -26months post partum. However, there is no information to calculate the date of delivery and the date of interview. 

Participants

3. Please clarify whether all women in the database were contacted to participate or a sub-sample. If a sub-sample – how were they selected?. If not able to be contacted, what are the characteristics of those who could not be contacted. Please address the question of selection bias thoroughly.

Response

We attempted to contact all women in the database. However, the several of the mothers were unreachable through their mobile phones. Identifying information were kept separately from the baseline survey. We used the patients register and mobile phone numbers we obtained during the baseline survey for the follow up. The ethical approval did not allow us to keep identifying information in the database. As such, we are unable to match the baseline data to the follow up data. We have addressed this as a potential limitation to our study. 

4. There is no sample size calculation – please include or explain why not?

Response 

We have included a sample size calculation.

Measures

5. Rather than describing the questions, please include them verbatim. Also, I counted only 5 question descriptions .

Response

Done 

6. Have the questions been validated against other measure of adherence that are not self-report or are they ‘de novo’– please give details and explanations

Response

The questions used to measure adherence in this study are pre-validated. Most evidence indicates that self-report adherence measures show moderate correspondence to other adherence measures and can significantly predict clinical outcomes.

Statistical analysis

7. The first 2 sentences fit better into the results. Also include characteristics of those not included in the 509 as well as the 10/509 who did not answer the adherence questions – and give reasons why they didn’t

Response

We have moved the sentences to the results section. 

Para 2

8. Clarify the outcome – “perfect adherence”

Response

We changed this to complete adherence and have defined this in the introduction.

9. Typo “A 95% confidence intervals..” – please correct

Response

Done

Results

1. Para 1“Receives” s/b received

Response

Changed to “received”.

2. Please add HIV duration and categories of months/years duration of HIV infection eg. 1.5-2yrs, 2-5 yrs, >5 - or some other meaningful division and include in the Tables , and comment on the result in the discussion. The same for months postpartum

3. Only keep this text if Table 1 is removed. Otherwise it is just a repeat

Response

We have merged Table 1 and Table 2.

4. Postpartum adherence: Specify that you mean the overall perfect adherence rate in sentence 1 and thereafter wherever this is the case. Or define adherence earlier in the text to mean perfect and the be consistent throughout the paper only using the word adherence

Response

Done

5. Multivariable findings: Change to Multivariable analyses

Response 

Done 

6. Remove the word “at least” as it is inaccurate, and cite the 95% CI

Response

Done

Tables

1. Table 1 is unnecessary as the information can be incorporated as a column with n (%) in

Response

We have merged Table 1 and Table 2. 

2. Table 2: It would strengthen the paper if you divide suboptimal adherence into at least 2 categories – based on the score. The ORs can compare perfect to suboptimal adherence overall, but this additional information is worth informing the reader

Response

We considered this and opted not to because the proportion of participants in the suboptimal category is not large. Further categorizing that group means we will loose statistical power. Also, there is no clinical relevance to do this. People with suboptimal adherence are likely to develop resistance and are to be targeted with interventions. 

3. Add footnotes to Table 3 to explain the *, **, ***

Response

Done

Discussion

1. Para 1 last sentence doesn’t make sense if you include “likely”. Please be specific

Response

The word “likely” has been deleted

2. Para 2.Explain why you assumed breast feeding will influence adherence – perhaps in the introduction and refer back to that with clear reasoning

Response

The assumption is based on a study that shows that women with older children, most of whom stopped breastfeeding by 13–18 months, reported more barriers and missed doses than women with younger children. We have stated this in both the introduction and the discussion.

3. Par 3: Within this paragraph, you give details regarding decline – why not give these in Para1 and link these ideas

Response

We thank the reviewer for this suggestion. We have implemented this suggestion.

4. Study limitations: Replace sentence 1 with “This study has some limitations”

Response 

Done

References:

1. Reference 17 is confusing – last author JJBid? no journal stated

 Response

Corrected

---

## [Decision Letter · Decision Letter 1]

11 Feb 2020

Levels and determinants of postpartum adherence to Antiretroviral Therapy in the Eastern Cape, South Africa

PONE-D-19-31896R1

Dear Dr. Ajayi,

We are pleased to inform you that your manuscript has been judged scientifically suitable for publication and will be formally accepted for publication once it complies with all outstanding technical requirements.

With kind regards,

Tanya Doherty, PhD

Academic Editor

PLOS ONE

Additional Editor Comments (optional):

Reviewers' comments:

Reviewer's Responses to Questions

**Comments to the Author**

1. If the authors have adequately addressed your comments raised in a previous round of review and you feel that this manuscript is now acceptable for publication, you may indicate that here to bypass the “Comments to the Author” section, enter your conflict of interest statement in the “Confidential to Editor” section, and submit your "Accept" recommendation.

Reviewer #1: All comments have been addressed

Reviewer #2: All comments have been addressed

2. Is the manuscript technically sound, and do the data support the conclusions?

Reviewer #1: Yes

Reviewer #2: (No Response)

3. Has the statistical analysis been performed appropriately and rigorously? 

Reviewer #1: Yes

Reviewer #2: (No Response)

4. Have the authors made all data underlying the findings in their manuscript fully available?

Reviewer #1: Yes

Reviewer #2: (No Response)

5. Is the manuscript presented in an intelligible fashion and written in standard English?

Reviewer #1: Yes

Reviewer #2: (No Response)

6. Review Comments to the Author

Reviewer #1: All my comments have been addressed satisfactorily. The authors have also included additional discussion points which I suggested.

Reviewer #2: (No Response)

7. PLOS authors have the option to publish the peer review history of their article (what does this mean?). If published, this will include your full peer review and any attached files.

Reviewer #1: No

Reviewer #2: No

---

## [Editor Report · Acceptance letter]

12 Feb 2020

PONE-D-19-31896R1 

Level and determinants of postpartum adherence to Antiretroviral Therapy in the Eastern Cape, South Africa 

Dear Dr. Ajayi:

I am pleased to inform you that your manuscript has been deemed suitable for publication in PLOS ONE. Congratulations! Your manuscript is now with our production department. 

With kind regards,

on behalf of

Professor Tanya Doherty 

Academic Editor

PLOS ONE